# Electric field control of radiative heat transfer in a superconducting circuit

Olivier Maillet [1✉], Diego Subero[1], Joonas T. Peltonen [1], Dmitry S. Golubev[1] & Jukka P. Pekola[1]

Heat is detrimental for the operation of quantum systems, yet it fundamentally behaves according to quantum mechanics, being phase coherent and universally quantum-limited regardless of its carriers. Due to their robustness, superconducting circuits integrating dissipative elements are ideal candidates to emulate many-body phenomena in quantum heat transport, hitherto scarcely explored experimentally. However, their ability to tackle the underlying full physical richness is severely hindered by the exclusive use of a magnetic flux as a control parameter and requires complementary approaches. Here, we introduce a dual, magnetic field-free circuit where charge quantization in a superconducting island enables thorough electric field control. We thus tune the thermal conductance, close to its quantum limit, of a single photonic channel between two mesoscopic reservoirs. We observe heat flow oscillations originating from the competition between Cooper-pair tunnelling and Coulomb repulsion in the island, well captured by a simple model. Our results highlight the consequences of charge-phase conjugation on heat transport, with promising applications in thermal management of quantum devices and design of microbolometers.

[1] QTF Centre of Excellence, Department of Applied Physics, Aalto University School of Science, P.O. Box 13500, 00076 Aalto, Finland. ✉email: olivier.maillet@aalto.fi

Ohmic resistors embedded in mesoscopic superconducting circuits are well suited to the study of radiative transfer physics, due to the correspondence between Planck's black-body radiation and Johnson–Nyquist noise originating from a resistive element[1,2]. Consider an arbitrary electrical circuit connecting two resistors $R_1$, $R_2$ kept at different temperatures $T_1$, $T_2$. Their voltage noises, which simply arise from black-body photon emission/absorption events, result in a global noise current flowing in the circuit, leading to Joule dissipation by the resistive elements. In the lumped element approximation, valid at low temperatures in the case of a millimeter scale circuit such as the one depicted in Fig. 1a, the thermal photon wavelength $hc/k_B T \approx 10$ cm at 150 mK is bigger than the typical size of the circuit. Thus the power transmission coefficient $\tau$ between the two resistances can be made explicit using a standard circuit approach[3,4]: $\tau(\omega) = R_1 R_2 / |Z_{\text{tot}}(\omega)|^2$, where $Z_{\text{tot}}(\omega)$ is the total circuit series impedance at angular frequency $\omega$. The net power $\dot{Q}_\gamma$ radiated from the hot to the cold resistor writes:

$$\dot{Q}_\gamma = \int_0^\infty \frac{d\omega}{2\pi} \tau(\omega) \hbar \omega [n_1(\omega) - n_2(\omega)]. \quad (1)$$

Here, $n_i(\omega) = 1/[\exp(\hbar\omega/k_B T_i) - 1]$ is the thermal population of the reservoir $i$, i.e., its Bose distribution at temperature $T_i$. The populations determine the thermal cutoff frequency $k_B T/\hbar$ of the radiation spectrum, which lies in the microwave range at cryogenic temperatures (~3 GHz at 150 mK). The maximum unity transmission leads to heat transfer at the universal quantum limit of thermal conductance $G_Q = \pi k_B^2 T / 6\hbar \approx (1\,\text{pW/K}^2) T$[5–13]. In our electrical approach (which may be generalized to arbitrary carriers statistics within the Landauer formalism[14]) this limit corresponds to perfect impedance matching, i.e., $R_1 = R_2$ with no additional contributions over the full black-body spectral range. Adding an appropriate tunable series reactance (a heat valve) permits tuning of the transmission coefficient without adding dissipation. Up to now, theoretical proposals[15] and

realizations[7,16,17] of a photonic heat valve only considered magnetic control, which is usually rather unpractical to implement. Besides, a larger degree of control may be required for fundamental investigations of heat transport in the quantum regime[18–21], e.g., by manipulating simultaneously charge and flux degrees of freedom. By contrast, electric control is now well established in electronic heat transport and thermoelectricity experiments, whether using a single-electron transistor[22], a quantum dot[23–28], or a quantum point contact[8,10,29–31], but it has not been considered for radiative heat transport. In this article, we experimentally demonstrate a fully electrostatic photonic heat valve operating close the quantum limit: in between our two thermal baths connected by superconducting lines, we include a Cooper-pair transistor (CPT)[32], a small superconducting island where electrostatic fields impede the charge transfer, a phenomenon commonly referred to as Coulomb blockade. Its magnitude can be simply adjusted by controlling the offset charge of the island with an electrostatic field via a nearby gate electrode. By varying the gate charge by an amount $e$, the effective series impedance is tuned from, ideally, matched case to mismatch, which in turn opens or closes the heat valve at will, as schematically displayed in Fig. 1b.

## Results

**Experimental setup and principle.** The sample (see Fig. 1a for a micrograph and Fig. 1c for the equivalent circuit) is measured in a dilution refrigerator, and addressed with filtered lines to minimize external noise. The system is a series combination of two nominally identical, small ($100 \times 100$ nm$^2$) Josephson junctions delimiting a small island of dimensions $1.4\,\mu\text{m} \times 170$ nm $\times 22$ nm with capacitance $C$, forming the CPT. This ensemble is attached on both sides to nominally identical thin copper films of volume $\Omega = 10\,\mu\text{m} \times 200$ nm $\times 12$ nm chosen so as to maximize their resistance (and thus the transmission factor $\tau$) while having minimal stray capacitance and thermal gradients. These resistors

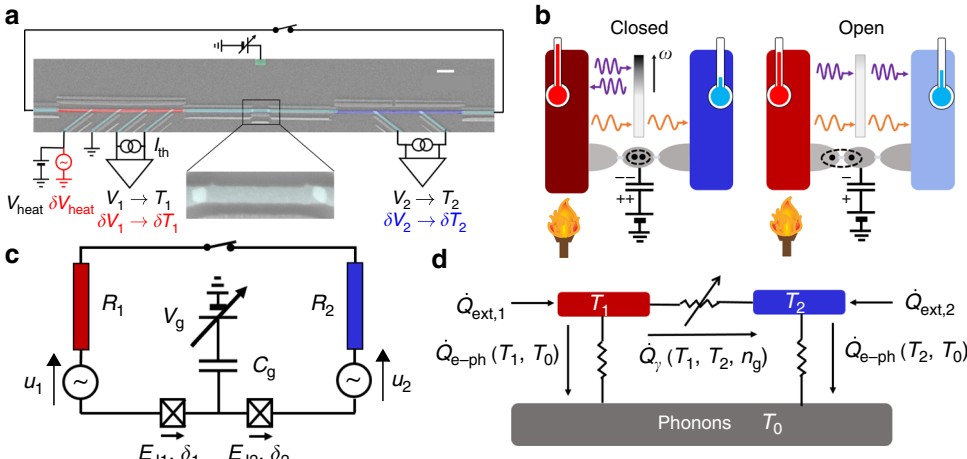

**Fig. 1 Experimental setup and principle of the electrostatic photonic heat valve. a** Scanning electron micrograph (scale bar: 2 μm) of the Cooper-pair transistor (central part with close-up), controlled with a side gate electrode (upper green lead) and connected via clean contacts and superconducting aluminium strips (light blue) to copper thin films (red and blue). Aluminium leads (oblique, light blue) are connected via oxide tunnel barriers to the films for local Joule heating (AC $\delta V_{\text{heat}}$ and DC $V_{\text{heat}}$) and/or electronic thermometry (using DC floating current sources and voltage amplifiers (resp. lock-in amplifiers) for DC ($V_{1,2}$) [resp. AC ($\delta V_{1,2}$)] read-out). A millimetric aluminium bonding wire closes the circuit, and can be removed to perform characterization and control measurements, hence the switch representation. **b** Principle of the experiment: the resistors are kept under constant temperature gradient, and emit Johnson noise. The CPT acts as a filter for radiated photons, with a frequency-dependent transmission coefficient schematically represented as a color gradient where the darker, the less transparent. In closed position, a single Cooper pair is localized in the superconducting island by applying an induced charge of $2e$ through the gate voltage, which reduces the noise current and hence the heat transfer. In open position, the $1e$ induced charge favors Cooper-pair tunneling and therefore increases the bandwidth of noise current, leading to increased heat transfer. **c** Electrical representation of the central circuit, with Johnson–Nyquist voltage noises $u_{1,2}$ represented as sources in series with the resistors $R_{1,2}$, $\delta_{1,2}$ the phases across the two Josephson junctions and $E_{J1,2}$ their Josephson energies. **d** Schematic representation of the thermal balance of the system.

thus act as quasiparticle filters for the CPT[32] as well as thermal baths, referred to in the following as source and drain. The clean electrical contact between the superconducting ($S$) circuit line and the normal metal ($N$) resistor acts as an Andreev mirror[33], transmitting charge but preventing heat carried by quasiparticles from flowing outside the resistors. As a result, electronic heat transport by quasiparticles is efficiently suppressed along the superconducting line at dilution temperatures[9]. The ensemble is electrically closed into a loop by a short (~5 mm) Al bonding wire to ensure that noise current carrying the photonic heat does flow and remain integrally in the so formed floating circuit[9]. Super-conducting leads are connected to the resistors through thin oxide barriers. These normal-insulator-superconductor (NIS) tunnel junctions enable one to measure the quasi-equilibrium electronic temperature of the resistor or to locally tune it via Joule heating[34]. Transport measurements made in a run prior to closing the loop (see Supplementary Methods) yield the resistances values $R_i \approx 290 \pm 30~\Omega$, the gate capacitance $C_g \approx 12$ aF, CPT (single electron) charging energy $E_c = e^2/2C = k_B \times 0.64$ K and Josephson energy per junction $E_J = k_B \times 0.69$ K $\sim E_c$.

We then investigate heat transport under an imposed temperature difference between the source and the drain. Any heat load brought externally to one of the resistors, say, the source, heats it up quickly (~1 ns) via electron–electron scattering to a quasi-equilibrium electronic temperature[35], whose steady-state value is determined by taking into account two inelastic scattering mechanisms for heated electrons. The first one is electron–phonon relaxation[36], which is minimized due to operation at low temperature and the small volume of the copper film. The second relaxation process occurs via electron–photon coupling[3] and is expected to be dominant at low temperature since electron–phonon thermal conductance vanishes as $G_{e-ph} \propto \Omega T^4$[34]. No intentional heat load is brought to the drain, and hence we can ascribe any temperature change observed there to a reservoir–reservoir heat flow, through the photonic channel.

A diagram summarizing the different heat flows in the system is depicted in Fig. 1d. The thermal balance of the system in steady state for a cryostat temperature $T_0$ writes for each resistor $i$:

$$\dot{Q}_{\text{ext},i} = \dot{Q}_{e-ph,i}(T_i, T_0) - (-1)^i \dot{Q}_\gamma(T_1, T_2, n_g), \quad (2)$$

where the electron–phonon heat flow for resistor $i$ is $\dot{Q}_{e-ph,i} = \Sigma\Omega(T_i^5 - T_0^5)$, with $\Sigma \approx 3.7 \times 10^9$ W m$^{-3}$ K$^{-5}$ the electron–phonon coupling constant, measured independently (see Supplementary Methods), for copper, and $\dot{Q}_\gamma$ is the source-drain heat flow. Using a lock-in amplifier, we measure small variations of temperature of peak amplitude $\delta T_{1,2}$ in both reservoirs upon a small sinusoidal heating at frequency $f = 77$ Hz added to the DC power brought through one source NIS junction. Assuming that steady-state is valid at each modulation increment ($f$ is much smaller than any relaxation rate) and $T_1 - T_2$, $\delta T_{1,2} \ll T_{1,2}$, from Eq. (2) we obtain an experimental value of thermal conductance between reservoirs 1 and 2 (see "Methods"):

$$G_\gamma = \frac{5\Sigma\Omega T_2^4}{\delta T_1/\delta T_2 - 1}, \quad (3)$$

with $T_2$ monitored within $\pm 1$ mK with a DC voltmeter. Such an AC technique allows us to measure heat currents with a resolution down to 100 aW Hz$^{-1/2}$, without suffering from excessive charge noise.

**Conductance modulation and model.** The temperature modulation amplitude in source and drain as a function of the applied gate voltage $V_g$ is shown in Fig. 2 for DC temperatures $T_1 = 203$ mK and $T_2 = 170$ mK. Clear oscillations are observed, that are $2e$-periodic in the gate charge $en_g = C_g V_g$. This is a strong

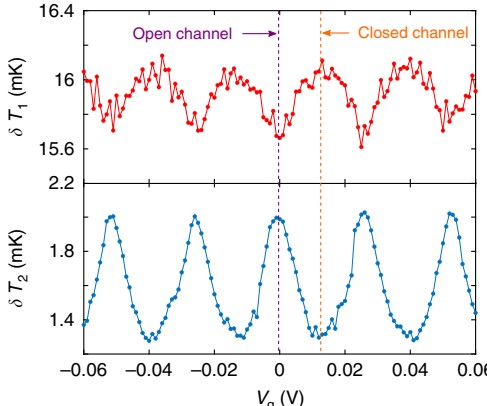

**Fig. 2 AC heating measurements.** Oscillation amplitude of the temperature response to small AC heating, recorded simultaneously in the source (upper panel) at DC temperature $T_1 = 203$ mK and drain (lower panel) at $T_2 = 170$ mK as a function of the applied gate voltage $V_g$.

indication that an interplay between Cooper-pair tunneling and Coulomb blockade in the superconducting island is behind the mechanism for heat modulation, similar to the critical current modulation of the CPT[32,37]. In addition, the temperature modulation is minimum (resp. maximum) in the source when that of the drain is maximum (resp. minimum), which can be correlated with opening (resp. closing) the photonic channel with the gate voltage. The amplitude of temperature oscillations is determined by the amount of power brought by the AC Joule heating signal, as well as the thermal balance: at our operation temperatures, electron–phonon relaxation in the reservoir is comparable with or dominant over the electron–photon mechanism. Therefore, the applied AC signal must be large enough to observe a sizeable response with good signal to noise ratio both in source and drain and to observe the gate modulation. On the other hand, it must be small enough to remain within the linear regime and keep the experimental definition of the photonic thermal conductance (3) valid. The data are taken with AC excitations kept in a range that satisfy both requirements.

Oscillations of the source-drain thermal conductance for three mean temperatures are represented in Fig. 3a, normalized to the thermal conductance quantum $G_Q$. The typical values are smaller than the conductance quantum $G_Q$ (~35% of the quantum limit at maximum), while the maximum achieved contrast $\mathcal{C} = (G_{\gamma,\max} - G_{\gamma,\min})/(G_{\gamma,\max} + G_{\gamma,\min}) = 0.28$ is far from reaching 1, as expected from the impedance mismatch introduced by the Josephson device. The thermal conductance oscillations can be understood in terms of $2e$ quantization of the charge on the island, jointly with the charge-phase conjugation at work in the CPT[32,38]. Assuming for simplicity that the two junctions are identical and neglecting quasiparticle excitations[32], the Hamiltonian $\hat{\mathcal{H}}$ of the system writes:

$$\hat{\mathcal{H}} = E_c(\hat{n} - n_g)^2 - 2E_J \cos\hat{\phi} \cos\frac{\delta}{2}, \quad (4)$$

where $\hat{n}$ is the number operator of excess paired electrons in the island, $\hat{\phi} = \delta_2 - \delta_1$ the phase of the island, and $\delta = \delta_1 + \delta_2$ the total phase across the CPT. $\hat{n}$ and $\hat{\phi}$ are canonically conjugated variables whose uncertainties $\Delta n$ and $\Delta\phi$ satisfy the relation $\Delta n \Delta\phi \geq 1$[38]. For odd values of $n_g$ the Coulomb gap $\Delta E = 4E_c(1 - n_g \bmod 2)$, which represents the electrostatic energy cost of adding one Cooper pair on the island, is closed. This leads to maximum quantum fluctuations of the charge degree of freedom $\hat{n}$ since Josephson coupling fixes the phase

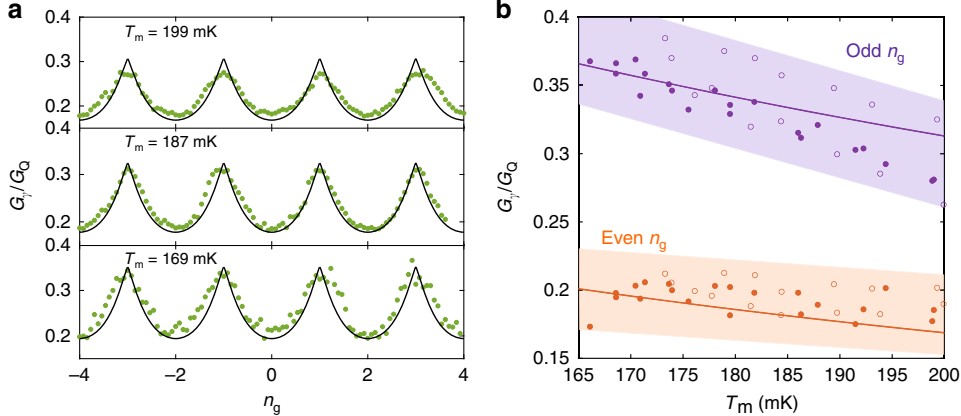

**Fig. 3 Coulomb oscillations of photonic thermal conductance and even-odd discrepancy. a** Source-drain thermal conductance, normalized to the quantum limit $G_Q = \pi k_B^2 T_m / 6\hbar$, as a function of the gate charge $n_g = C_g V_g / e$, for three median temperatures ($T_m = 169, 187$ and $199$ mK). The theoretical lines are the application of Eq. (5). **b** Normalized conductances measured at gate open (purple dots) and closed (orange dots) positions, at cryostat temperatures 100 mK (full dots) and 150 mK (empty dots). The solid lines are the application of Eq. (5) for $n_g = 0 \bmod 2$ (orange) and $n_g = 1 \bmod 2$ (purple). The colored zones delimit the typical uncertainty, based on the measurement noise and parameters uncertainties.

across the junctions and thus $\hat{\phi}$. Therefore, the Josephson supercurrent flowing across the CPT is maximum, which in turn minimizes the Josephson inductance $L_J = \hbar/2eI_C$, where $I_C$ is the CPT critical current. As a result, the bandwidth for thermal excitations is increased and so is the thermal conductance. Conversely, for even values of the gate charge, the Coulomb gap is maximized, which tends to freeze the number of Cooper pairs on the island. As a result, quantum phase fluctuations are increased, thus leading to a reduction of the effective Josephson coupling and thus of the cutoff frequency for thermal currents.

The above picture can be captured by a simple circuit model. The Josephson element under zero bias, small junction capacitance and in a low impedance environment ($R_{1,2} \ll R_Q = h/4e^2 = 6.45$ kΩ) may be approximated as a gate-tunable inductor $L_J$ in series with the resistors in the frequency range relevant for thermal photons at 100–200 mK (see supplementary information for detailed calculations and modelling). The circuit with series impedance $Z_{tot}(\omega) = R_1 + R_2 + iL_J\omega$ is thus a low-pass filter for thermal radiation with a gate-tunable cutoff frequency $\omega_c(n_g) = (R_1 + R_2)/L_J(n_g)$. The gate dependence of the critical current $I_C$ is derived by finding the maximum tilt, allowed in the supercurrent branch, of the effective Josephson washboard potential modified by Coulomb interactions on the island, $I_C(n_g) = 2e/\hbar \max_\delta \partial E_0(n_g, \delta)/\partial\delta$, where $E_0(n_g, \delta)$ is the ground eigenenergy band of the system (see Supplementary Note 1) obtained from ((4)). We can calculate thereafter the theoretical photonic heat conductance in the small temperature difference limit, $G_\gamma = \dot{Q}_\gamma/(T_1 - T_2)$. With $T_m = (T_1 + T_2)/2$ the mean temperature and from the heat flow expression ((1)), we obtain:

$$G_\gamma(n_g, T_m) = \frac{2k_B^2 T_m R_1 R_2}{\pi\hbar(R_1 + R_2)^2} \times \int_0^\infty \frac{x^2 e^x \mathrm{d}x}{(e^x - 1)^2} \frac{1}{1 + x^2/x_c^2(n_g)}. \quad (5)$$

Here, $x_c(n_g) = \hbar\omega_c(n_g)/k_B T_m$ is the reduced circuit cutoff frequency. Incidentally, in our mismatched situation this parameter introduces an additional dependence in temperature, leading to a departure from the simple $G_\gamma \propto T_m$ picture[7]. In Fig. 3a we see that despite its simplicity, our model reproduces well the main features of our experimental data, with essentially no free parameters. The thermal conductances at odd and even $n_g$ are shown in Fig. 3b, for two cryostat temperatures $T_0 = 100$ and 150 mK, as a function of the mean electron temperature $T_m$, again in good agreement with the model. A refined model may include e.g., anharmonicity, phase diffusion, the junction asymmetry, a finite stray inductance, as well as quasiparticle poisoning[37,39] (see Supplementary Discussion). In addition, for large source temperatures, the gradient becomes large enough for the thermal conductance to be ill-defined.

## Discussion

The performance of the device, which encompasses both the contrast in Coulomb oscillations and the maximum achieved heat flow, may be condensed in the coefficient $\beta = \mathcal{C} \times G_{\gamma,max}/G_Q$, where 1 indicates maximal performance. A device with $\beta = 1$ would have a low-pass cutoff frequency $\omega_c \gg k_B T/\hbar$ in open position (fully quantum-limited heat conduction) and $\omega_c \ll k_B T/\hbar$ in closed position (fully filtered thermal noise). $\beta$ is rather small in our experiment (at best 0.1) but may be improved with optimized device parameters for e.g., microbolometry[40–42] or refrigeration[43] purposes. For instance, the ratio $E_J/E_c$ that determines $G_{\gamma,min}$ may be reduced by decreasing the junctions size, while the negative impact of the subsequently reduced $E_J$ on $G_{\gamma,max}$ may be compensated by designing more resistive metallic baths in order to improve impedance matching, as long as $R_{1,2} \ll R_Q$.

Our experiment establishes that the electron–photon relaxation mechanism can be controlled with electric field, down to a single electric charge level, in a dual manner to magnetic field control down to a single flux quantum. This could allow for instance sensitive thermal charge detection with minimal back-action from a temperature (rather than voltage) biased electrometer. Note that the recent demonstration of reversible gate-controlled suppression of supercurrent in conventional superconducting nano-constrictions[44–46] may be conveniently employed for photonic heat control as well. Here, despite the fundamentally different possible microscopic mechanisms at work, the constriction could advantageously replace the CPT, reducing the fabrication complexity. More fundamentally, a natural development would be to associate magnetic and electric field control on the same circuit to explore many-body effects due to e.g., a high-impedance environment[19,47] on heat transport, which should lead to nontrivial thermal conductance laws. In addition, the finite frequency content of noise exchange between the resistors is not addressed: a suitable setup would allow to monitor nonequilibrium voltage

fluctuations[48] at both resistors' ends. This could extend to the quantum regime investigations of entropy production by a heat flow[48,49].

## Methods

**Fabrication and setup**. All the junctions, contacts and leads were fabricated in a single-electron-beam lithography step using the Dolan bridge technique[50]. A silicon wafer with 100 nm grown silicon oxide was coated with a stack of a 1 μm thick layer of poly(methylmetalcrylate–methacrylic) acid P(MMA–MAA) resist spun for 60 s at 4000 rpm and baked at 160 °C in three steps and on top of it a 100 nm thick layer of polymethyl-metacrylate (PMMA) spun for 60 s at 4000 rpm and baked at 160 °C. The samples were patterned with electron-beam lithography and subsequently developed using methyl isobutyl ketone (MIBK 1:3 Isopropanol) for PMMA and methylglycol methanol (1:2) to create the undercut in the MAA resist. The metallic parts were evaporated in three steps in the following order: Al, Al, Cu, with an in situ oxidation step under low oxygen pressure in between the two first steps to create the tunnel barriers for both NIS probes and Josephson junctions. The clean contacts necessary for a lossless transmission between the normal and superconducting parts were created through the second and third evaporation steps. The resist was then removed using hot acetone.

The sample was mounted in a stage with double brass enclosure acting as a radiation shield. The stage was thermally anchored to the mixing chamber of a small homemade dilution refrigerator with 50 mK base temperature. All lines were filtered with standard lossy coaxial cables with bandwidth 0–~0 kHz. Amplification of the output voltage signals at the ends of the NIS probes was realized using a room temperature low noise voltage amplifier Femto DLVPA-100-F-D. The DC signals were applied and read using standard programmable sources and multimeters. The effective integration bandwidth around the oscillator frequency for AC measurements was 0.26 Hz. The calibration of the local electronic thermometers was done by monitoring the voltage drop at the ends of the current-biased ($I$th ≈ 160 pA) SINIS configuration while ramping up the cryostat temperature up to 350 mK (for more details see e.g., ref. [34]).

**Thermal conductance measurement**. A precise observation of the heat flow modulation and averaging of even a moderate number of datasets is made difficult in a pure DC measurement of the electron temperature by unavoidable charge noise that manifests in single electron devices[51] when using long measurement times. Nevertheless, the DC values are recorded as a reference throughout the gate sweep with a typical uncertainty of ±1 mK for a ~1 s averaging time, which is too large for a straight DC measurement where gate modulation depths are of this order but very small when measuring the thermal conductance with the lock-in technique (see below).

The heat balance equations are written in the main text. To measure the heat conductance we impose a small, AC heating signal on top of the DC one that establishes the thermal gradient. The AC frequency $f$ ~77 Hz is small enough, on the one hand, for the quasi-equilibrium temperature of the electron gas to be defined (the electron–electron scattering time $\tau_{e-e}$ ~ 1 ns $\ll f^{-1}$[35]), and for the steady-state hypothesis to be valid at any relevant measurement timescale on the other hand: indeed, the typical energy relaxation timescale is upper bounded by the electron–phonon relaxation time $\tau_{e-ph}$ ~10–100 μs at 100 mK in copper[52], which is much shorter than the typical AC modulation timescale $f^{-1}$. Therefore the power balance equation written in the main text can be re-written for a steady state displaced from $(T_1, T_2)$ to $(T_1 + \delta T_1, T_2 + \delta T_2)$, and expanded at first order in the increments $\delta T_{1,2} \ll T_{1,2}$, assuming the phonon temperature $T_0$ (taken equal to the cryostat temperature), remains constant:

$$\dot{Q}_{ext,2} \approx \dot{Q}_{e-ph,2}(T_2, T_0) + 5\Sigma\Omega T_2^4 \delta T_2$$
$$- \left[\dot{Q}_\gamma(T_1, T_2) + \frac{\partial \dot{Q}_\gamma}{\partial T_1}\bigg|_{T_1} \delta T_1 + \frac{\partial \dot{Q}_\gamma}{\partial T_2}\bigg|_{T_2} \delta T_2\right]. \quad (6)$$

There we identify the power balance terms for steady-state $(T_1, T_2)$ which cancel out. Noticing that $T_1 - T_2 \ll T_{1,2,m}$, and disregarding thermal rectification phenomena[53] (the couplings to the resistances are nominally identical), we can make the following approximation that defines the experimental thermal conductance at the mean temperature $T_m$:

$$G_\gamma(T_m) \equiv \frac{\partial \dot{Q}_\gamma}{\partial T_2}\bigg|_{T_2 \to T_m} \approx -\frac{\partial \dot{Q}_\gamma}{\partial T_1}\bigg|_{T_1 \to T_m}, \quad (7)$$

with corrections up to a factor $(T_1 - T_2)/2T_m$, which become important for gradients larger than ~50 mK, limiting the applicability of the method to roughly a source temperature of 230 mK. Thanks to the linearity, under these conditions, of Eq. (6), we can replace the increments by their RMS value measured with a lock-in amplifier. Keeping for them the same notation and rearranging the terms in Eq. (6), we finally obtain the value of thermal conductance extracted from lock-in measurements and used in the main text. Note that given our low modulation frequency we expect and indeed observe a negligible quadrature response of the lock-in amplifier read-out for a 0° phase reference. Such a response should be significant only at AC heating frequencies comparable with or higher than the electron–photon or electron–phonon relaxation rates[36] (hence at kHz frequencies or above), but may also be visible at lower frequencies due to spurious capacitive cross-talk in the AC line which increases upon increasing the heating signal frequency.

## Data availability

The data that support the findings of this study are available from the corresponding author upon reasonable request.

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

## Acknowledgements
We thank E. Aurell, F. Blanchet, A. Gubaydullin, E. Mannila, Yu. Pashkin, and K. Saito for helpful discussions. This work was performed as part of the Academy of Finland Centre of Excellence program (project 310257). We acknowledge the provision of facilities by Aalto University at OtaNano-Micronova Nanofabrication Centre.

## Author contributions
The experiment was conceived by O.M., D.S.G., and J.P.P and performed by O.M., with contributions from D.S. and technical support provided by J.T.P.; O.M. fabricated the sample; the model was developed by O.M. and discussed with D.S.G. and J.P.P.; the results were analyzed and the manuscript written by O.M. with contributions from all authors.

## Competing interests
The authors declare no competing interests.
