## [Peer Review File · Nature Communications]

REVIEWER COMMENTS

Reviewer #1 (Remarks to the Author):

This paper reports a realization of purely electrostatic photonic heat valve in a superconducting circuit, where the thermal conductance of a single photonic channel between two reservoirs can be tuned close to its quantum limit. The observed temperature modulation and hence that of the thermal conductance show $2e$ -periodicity as a function of the gate charge up to a certain temperature limit below which the quasiparticle poisoning can be neglected. The authors explain the observed heat flow oscillations due to the competition between Cooper-pair tunneling and Coulomb repulsion in a gate-controlled Cooper pair transistor with a simple model. By neglecting the junction asymmetry and quasiparticle contributions and by approximating the Josephson junction as a gate-tunable LC resonator, the authors explain their measurements with no free parameters simply based on the Landauer scattering theory picture and the heat balance equation incorporating the phonon bath. They also justify their approximations in the Supplementary Information. As the gate charge can be varied down to a single charge level, the heat valve can be closed or opened at will paving the way for the magnetic-field-free control of electron-photon relaxation mechanisms.

I find the presentation of the manuscript succinct, informative and well-written. The motivation, experimental details and the theoretical description to support their observation are explicit and nicely presented throughout the paper. The experimental and theoretical methodologies seem pertinent and sound. The expected ramification of this work is also summarized in the concluding remarks followed by the Methods section which details the fabrication of the device and the analysis in the paper. The Supplementary Information further addresses various details of the experiment and the theory. I think that this paper will be influential in several cutting edge fields of nano science in strong relevance to quantum thermodynamics, superconducting coherent caloritronics and quantum thermoelectrics. In view of the above points, I would recommend this manuscript for publication in Nature Communications.

Nevertheless, I wonder if there were any precedent theoretical and/or experimental works related to the authors' findings regarding the electric field control of heat even if they may not be of photonic nature or might have been studied in different systems, as the references are completely lacking in the present manuscript. It will be helpful for the readers if a brief sketch of the previous efforts can be provided on which the present work is based. In this way, I believe that the significance of the authors' work filling the preexisting gaps will be more clearly visible.

Reviewer #2 (Remarks to the Author):

The authors report a gate-modulated thermal conductivity between two reservoirs using a Cooper pair transistor as a valve. This is an interesting experiment and, I think with some discussion & possible modification, it could be suitable for publication in Nature Communications.

I read and re-read parts of this manuscript to try to re-cast it into a picture that is closer to how I think about these things. Part of why I felt the need to do this is that it seems the manuscript is written to make this sound maybe a little more profound than I think it needs to, or maybe should be. It could also be that I really am failing to grasp what this is showing but, I think I understand enough to ask some basic questions:

1. Equation 1 indicates that heat flow can be cast in a Landauer picture, as the integral of the

difference of the equilibrium mode occupancies of two reservoirs, weighted by a single-channel transmission factor $\tau(\omega)$.

Invoking Landauer here, is, I guess, a way to connect with a mesoscopic picture of transport, but I really don't think that it's necessary. They write down an expression for what this transmission coefficient is, $\tau(\omega) = 4 R_1 R_2 / |Z_{\text{tot}}(\omega)|^2$. In their circuit model, this would correspond to what you would derive for the rf/microwave transmission coefficient with port impedances R_1 and R_2 . As such, what they are saying, and what I think the paper is about, is really how 1D black body radiation propagates through a gate-tunable filter, the Cooper pair transistor. This heat transport problem requires integrating over a broad power spectral density that is mostly thermal up to the 'thermal bandwidth' cutoff, $k_B T / \hbar$, rather than a narrow bandwidth centered somewhere in the quantum limit $\omega \gg k_B T / \hbar$.

This is all a long-winded way to ask whether the authors have purposely chosen to not discuss this as a 1D blackbody + filter picture explicitly. It seems much simpler to frame the discussion this way, rather than as about approaching a quantum of thermal conductance or invoking Landauer. Although the latter sounds more exciting to some people, I think it oversells this, especially when we're talking about one scattering element.

I will note that I sat down and estimated the transmission based on some simple assumptions to understand the series impedance of the Cooper pair transistor. Since the plasma frequency is probably very high (I estimate 30+ GHz), this should be well-approximated by the gate-modulated Josephson inductance. With nanoampere-scale critical currents (shown from the Joyez-type 3-state calculation in the Supplementary), the impedance looks to be something like a few hundred Ohms, modulating by a few hundred Ohms. However, just plugging into their equation for the transmission (again, just what one would calculate for in general for two different port impedances), I see a gate-modulated low-pass filter response with a 3 dB cutoff that varies between 1.5 to 3.5 GHz. When looked at in this way, it's pretty clear that heat conveyed via noise propagating at these frequencies will be modulated by scattering from the CPT impedance. I have attached a PDF of a Jupyter notebook showing what these back-of-the-envelope estimates look like to me. I hope that by including this, you can tell me more precisely whether I'm misinterpreting the basic picture.

2. Temperature scales, Other temperature choices

Apologies if I missed it, but I'm not sure what motivates the choice for the reservoir & cryostat temperatures. Presumably they could all be much lower. How does the contrast in the temperature oscillations vary with this temperature? Not in a theoretical sense, but in practice? The measurements in Fig 3b are taken at relatively high temperatures compared to the base temperature of their dilution refrigerator.

3. Quasiparticle poisoning

This is a much smaller comment, but important nonetheless. In the Supplementary, the authors discuss how they think about quasiparticle poisoning. It's a little bit of an outlier today as it judges the degree of poisoning from an averaged switching current. This is a very old way (Joyez thesis) to assess the parity. It gives some weighted measure of the average parity, but taking the average removes any hints of the fast dynamics that underlie all aluminum single-charge and even transmon devices. To be specific, even with a 2e periodic switching current curve, there can be significant quasiparticle poisoning. This is more clearly shown in histogrammed switching current statistics, shown in work from the mid 2000s on Cooper pair transistors from NIST, UNSW, and others. It seems that Joyez is the only reference for quasiparticles in these devices, although the literature is pretty

deep at this point. I believe the authors' own group has published other CPT work that is much better-referenced in this regard.

3. Device limitations/Improvements.

It would be nice if the authors could discuss how this device might be improved, or if its performance (contrast) can be improved. If this picture of gate-modulated low-pass filtering is ok, then one can imagine producing a device with more contrast in the minimum/maximum critical currents. One could also impedance match the lead resistances to the CPT impedance. There's some practical limit to this, but more generally, is there some practical benefit (research or applied) to improving the performance?

Summary

This is an interesting paper, although I think that some of the language is a little dramatic for my taste (although I acknowledge that some of this seems necessary for certain journals), and while I'm not suggesting it for publication right away, I'm hoping that my comments/questions can help clarify the picture in the paper so that it's more instructive to a wider audience. I think the bigger questions they're asking are interesting in general, but I think that the experimental discussion could be made clearer.

Reviewer #3 (Remarks to the Author):

In this work, Maillet et al present the realization of heat flow modulation using a fully electric knob, i.e., a Cooper pair transistor. The authors succeed in opening and closing a photonic heat channel between two normal metal reservoirs by playing with the Cooper pair tunneling and Coulomb blockade effect. To do so they use a nice lockin technique to monitor temperature variations, they demonstrate (supp material) that the photonic channel is the main path for heat transport and, finally, they demonstrate that temperature modulation is $2e$ -periodic, univocal proof of the relationship between heat flow and $2e$ quantization in the transistor. There is a sufficient amount of data presented along the paper, figures are extremely simple but really evoking of the phenomenon under investigation. Theoretical analysis of the results is also very clearly explained, rigorous and convincing. To summarize, the work presented by Maillet et al is very beautiful and interesting for a broad scientific community and I recommend its publication in Nat Commun.

In the following I pose very few questions/suggestions for the authors:

- 1) In the introduction (abstract) part the authors comment on the flux-charge duality. This statement is confusing to me as they want to avoid any reference to a magnetic flux controlling knob. Would it be better to talk about charge-phase conjugation?
- 2) Along the paper I didn't get what factors determine the amplitude of the temperature oscillations that the authors are observing. Is it possible to increase this amplitude by, e.g., increasing the thermal gradient or decreasing the cryostat base temperature?
- 3) I find that the definition of the maximum achieved contrast a bit confusing. Max contrast = 1 means simply that $G_{\gamma, \min} = 0$, am I right?
- 4) What do you mean by "negligible retarded response" (methods section) and how do you observe it experimentally?

5) What do you mean by "the issue of finite frequency behavior" (conclusions section)?

6) Finally, I have a fundamental question. What is exactly the role played by the superconductor? As far as I understand it, the role of the superconducting part is to block heat currents mediated by quasiparticles. If this is the case, could it be possible to demonstrate a similar effect using a conventional normal-metal single-electron transistor stacked between superconducting leads or am I missing something? Is that the equivalent effect that you are observing at large temperatures where you attribute the $1e$ -periodicity to quasiparticle poisoning (supp material)? If this is the case, has this effect been observed experimentally previously?

Reviewer #1 (Remarks to the Author):

This paper reports a realization of purely electrostatic photonic heat valve in a superconducting circuit, where the thermal conductance of a single photonic channel between two reservoirs can be tuned close to its quantum limit. The observed temperature modulation and hence that of the thermal conductance show $2e$ -periodicity as a function of the gate charge up to a certain temperature limit below which the quasiparticle poisoning can be neglected. The authors explain the observed heat flow oscillations due to the competition between Cooper-pair tunneling and Coulomb repulsion in a gate-controlled Cooper pair transistor with a simple model. By neglecting the junction asymmetry and quasiparticle contributions and by approximating the Josephson junction as a gate-tunable LC resonator, the authors explain their measurements with no free parameters simply based on the Landauer scattering theory picture and the heat balance equation incorporating the phonon bath. They also justify their approximations in the Supplementary Information. As the gate charge can be varied down to a single charge level, the heat valve can be closed or opened at will paving the way for the magnetic-field-free control of electron-photon relaxation mechanisms.

I find the presentation of the manuscript succinct, informative and well-written. The motivation, experimental details and the theoretical description to support their observation are explicit and nicely presented throughout the paper. The experimental and theoretical methodologies seem pertinent and sound. The expected ramification of this work is also summarized in the concluding remarks followed by the Methods section which details the fabrication of the device and the analysis in the paper. The Supplementary Information further addresses various details of the experiment and the theory. I think that this paper will be influential in several cutting edge fields of nano science in strong relevance to quantum thermodynamics, superconducting coherent caloritronics and quantum thermoelectrics. In view of the above points, I would recommend this manuscript for publication in Nature Communications.

We thank the referee for his/her positive evaluation of our work

Nevertheless, I wonder if there were any precedent theoretical and/or experimental works related to the authors' findings regarding the electric field control of heat even if they may not be of photonic nature or might have been studied in different systems, as the references are completely lacking in the present manuscript. It will be helpful for the readers if a brief sketch of the previous efforts can be provided on which the present work is based. In this way, I believe that the significance of the authors' work filling the preexisting gaps will be more clearly visible.

We agree with the referee that prior work deserves better mention and puts our results in a clearer context. To our knowledge, this is the first time the photon heat flow is controlled electrically in a heat transport experiment. Theoretically, the important work presented in Ref. [T. Ojanen & A.-P. Jauho PRL **100** 155902 (2008)] suggests a magnetic flux control but obviously also allows any kind of tunable effective impedance, including one tuned by a gate electrode. As for electric field control of heat mediated by electrons, in the context of quantum heat transport and heat engines, there are indeed

plenty of references from the past fifteen years, some of which being very close to our design, where gate control and QPC control is implemented.

Thus, to give a better account of the context and prior work upon which this work builds, we have added in the revised manuscript a paragraph in the introduction, before “In this Letter..”, that refers to those prior works. In addition, we open the final discussion with the possibility to use the recently demonstrated (see e.g. [G. De Simoni *et al.*, Nat. Nano. **13** 802-805 (2018)]) field effect control of the supercurrent in a superconducting constriction. Although the origin of such an effect seems still unclear, it has been reproduced by several experiments since then, and we believe that it can be conveniently used as a photon heat switch, since it avoids the need for oxidation and/or additional angle evaporation during the device fabrication.

The authors report a gate-modulated thermal conductivity between two reservoirs using a Cooper pair transistor as a valve. This is an interesting experiment and, I think with some discussion & possible modification, it could be suitable for publication in Nature Communications.

We thank the referee for his/her assessment of our work and his/her very careful and thorough review, which we think truly improved the paper.

I read and re-read parts of this manuscript to try to re-cast it into a picture that is closer to how I think about these things. Part of why I felt the need to do this is that it seems the manuscript is written to make this sound maybe a little more profound than I think it needs to, or maybe should be. It could also be that I really am failing to grasp what this is showing but, I think I understand enough to ask some basic questions:

1. Equation 1 indicates that heat flow can be cast in a Landauer picture, as the integral of the difference of the equilibrium mode occupancies of two reservoirs, weighted by a single-channel transmission factor $\tau(\omega)$.

Invoking Landauer here, is, I guess, a way to connect with a mesoscopic picture of transport, but I really don't think that it's necessary. They write down an expression for what this transmission coefficient is, $\tau(\omega) = 4 R_1 R_2 / |Z_{\text{tot}}(\omega)|^2$. In their circuit model, this would correspond to what you would derive for the rf/microwave transmission coefficient with port impedances R_1 and R_2 . As such, what they are saying, and what I think the paper is about, is really how 1D black body radiation propagates through a gate-tunable filter, the Cooper pair transistor. This heat transport problem requires integrating over a broad power spectral density that is mostly thermal up to the 'thermal bandwidth' cutoff, $k_B T / \hbar$, rather than a narrow bandwidth centered somewhere in the quantum limit $\omega \gg k_B T / \hbar$.

This is all a long-winded way to ask whether the authors have purposely chosen to not discuss this as a 1D blackbody + filter picture explicitly. It seems much simpler to frame the discussion this way, rather than as about approaching a quantum of thermal conductance or invoking Landauer. Although the latter sounds more exciting to some people, I think it oversells this, especially when we're talking about one scattering element.

The referee is absolutely correct: the expression for heat flow Eq. (1) may be directly derived from purely electrical arguments, following the "blackbody+filter" approach he/she introduces in the attachment. This is in fact also our way to practically deal with the analysis (see SuppMat). In that sense, the connection with the Landauer approach may indeed seem superfluous here, as we did not use it to derive Eq. (1) nor any other result. However, we stress that the main focus of our work is heat transport: from this point of view, Eq. (1) is valid beyond the particular circuit approach, regardless of heat carriers. Besides, the connection to the quantum of thermal conductance seems like an essential point to us, in particular since we discuss (see modifications) the valve performance, which involves this quantity. Nevertheless, we acknowledge that in the present form of the text, Eq. (1) looks as derived from a generic Landauer approach, which is misleading as it is not the case in our analysis. We can thus "tone down" significantly the emphasis on this aspect, and propose to introduce first the

circuit picture and only after that briefly mention its generalization to other heat transport situations within the Landauer formalism, without mentioning it further in the manuscript.

We have significantly re-written the introduction (see version with coloured modifications), leaving only a note about the Landauer approach, and suppressed any reference to it further in the text and in the SI.

On a side note, even in a “circuit” analysis of the electromagnetic heat transport issue, the expression of the transmission coefficient may not be always obtained by a “straightforward” scattering matrix picture. One may consider the experimentally relevant case of our circuit with resistors whose values are now comparable with the superconducting resistance quantum $h/4e^2 = 6.45 k\Omega$. In that situation, the Josephson element may not be easily approximated as an impedance because of environmental Coulomb blockade [G-L. Ingold & Yu. V. Nazarov, *Charge Tunneling Rates in Ultrasmall Junctions* (1992)], which leads to more complex and (in our view) richer physics also in the heat sector [G. Thomas *et al.*, PRB **100** 094508 (2019)]: the consequences of this phenomenon on heat transport have, up to date, not been experimentally investigated to our knowledge.

I will note that I sat down and estimated the transmission based on some simple assumptions to understand the series impedance of the Cooper pair transistor. Since the plasma frequency is probably very high (I estimate 30+ GHz), this should be well-approximated by the gate-modulated Josephson inductance. With nanoampere-scale critical currents (shown from the Joyez-type 3-state calculation in the Supplementary), the impedance looks to be something like a few hundred Ohms, modulating by a few hundred Ohms. However, just plugging into their equation for the transmission (again, just what one would calculate for in general for two different port impedances), I see a gate-modulated low-pass filter response with a 3 dB cutoff that varies between 1.5 to 3.5 GHz. When looked at in this way, it's pretty clear that heat conveyed via noise propagating at these frequencies will be modulated by scattering from the CPT impedance. I have attached a PDF of a Jupyter notebook showing what these back-of-the-envelope estimates look like to me. I hope that by including this, you can tell me more precisely whether I'm misinterpreting the basic picture.

We thank the referee for this observation and really appreciate the effort he/she made to perform calculations that support his/her (absolutely correct) claim. Indeed, as pointed out here and in the attachment, the plasma frequency is high enough to be irrelevant, and what really matters is the low-pass filter cut-off frequency imposed by $(R_1 + R_2)/L_J(n_g)$ (L_J is the Josephson inductance), which is also gate-tunable. In our analysis, putting C as C_Σ (as we wrote by mistake first, thanks for pointing this) or C_{series} (which is the correct value) or 0 indeed changes almost nothing to the thermal conductance, for the reason explained by the referee. In that light, the text is misleading, because we mention only the plasma frequency tunability. We thus agree with the referee that the explanation in the text can be very conveniently made simpler, in terms of the low-pass filtering only due to the tunable Josephson inductance. In that view, the theoretical expression of the thermal conductance, Eq. (6) in the main text, may also be simplified accordingly.

We have rewritten the modelling paragraph, before and after Eq. (6). It now makes no mention of the plasma frequency and makes the low-pass filtering picture explicit. The theoretical conductance Eq. (6) is also simplified following this picture. We have also re-written the corresponding SI section, which justifies the use of a pure inductor.

Regarding the comment in the Jupyter file about the comparison between thermal cut-off and circuit cut-off, we agree with the referee that a device with good contrast should have a circuit cut-off well

below the thermal one in closed position, and well above in open position if one wants a maximal, quantum-limited heat flow. Our device is indeed not optimal in that sense: the circuit cut-offs are in any case neither much larger nor much smaller than the thermal cut-off, and there is room for device optimization. We have added this to the main text in the newly added optimization paragraph (see related question below).

2. Temperature scales, Other temperature choices

Apologies if I missed it, but I'm not sure what motivates the choice for the reservoir & cryostat temperatures. Presumably they could all be much lower. How does the contrast in the temperature oscillations vary with this temperature? Not in a theoretical sense, but in practice? The measurements in Fig 3b are taken at relatively high temperatures compared to the base temperature of their dilution refrigerator.

The electron temperature in the resistors saturates near 150 mK in our sample, and we attribute this to external parasitic radiation leaking in the loop through the NIS junctions or the bonding wire closing the loop. Usual saturations happen around 100 mK, but here we have deliberately chosen to use smaller volume resistors to obtain a decent trade-off between large resistance values (i.e. more than 100Ω but less than $1 \text{ k}\Omega$ to avoid environmental back-action) and minimized impact of electron-phonon coupling, which increases linearly with the volume. As a result, the saturation temperature is higher. This limitation is of purely technical nature, and some optimization would make lower cryostat temperatures much more convenient indeed.

We stress that our lock-in measurement allows us to conveniently extract the pure photonic heat conductance (or generally, the direct reservoir-reservoir heat conductance) regardless of the other heat flows not linking the two reservoirs. However, as we increase the cryostat temperature, electron-phonon coupling becomes the dominant relaxation mechanism in each reservoir (in fact it is already seriously competing with photonic relaxation at 100-150 mK). As a result, for the same amount of AC heating power, a higher cryostat temperature leads to reduced temperature modulation amplitude and Coulomb oscillations contrast both in source and drain, which has consequences on the signal to noise ratio of the extracted thermal conductance. Therefore, for too high temperatures (in our case above 220-300 mK), the signal-to-noise ratio is too poor to extract the photonic heat conductance with reasonable accuracy.

We have added a paragraph before "Oscillations of the source-drain..." which details this issue, also addressed by Referee #3.

3. Quasiparticle poisoning

This is a much smaller comment, but important nonetheless. In the Supplementary, the authors discuss how they think about quasiparticle poisoning. It's a little bit of an outlier today as it judges the degree of poisoning from an averaged switching current. This is a very old way (Joyez thesis) to assess the parity. It gives some weighted measure of the average parity, but taking the average removes any hints of the fast dynamics that underlie all aluminum single-charge and even transmon devices. To be

specific, even with a $2e$ periodic switching current curve, there can be significant quasiparticle poisoning. This is more clearly shown in histogrammed switching current statistics, shown in work from the mid 2000s on Cooper pair transistors from NIST, UNSW, and others. It seems that Joyez is the only reference for quasiparticles in these devices, although the literature is pretty deep at this point. I believe the authors' own group has published other CPT work that is much better-referenced in this regard.

We agree with the referee that $2e$ parity on average does not equal a quasiparticle-free device, that this question deserves a better treatment and that more recent references are appropriate. We do have histograms of switching currents, an example of which we propose to add to the supplementary material, along with further references. While the estimates are rather rough (even in the best case, there is always a residual overlap between the histograms peaks corresponding to even and odd switching events), we observe that for an odd applied offset charge, up to roughly 200 mK, the device is free of unpaired quasiparticle excitations more than 96 % of the time at worst. This quickly changes near 200 mK as the even/odd free energy difference is decreased down to be comparable with the other energy scales of the device, such that it is even visible on averaged data. Using different ramp times we have estimated that the typical odd/even parity lifetimes below 200 mK are in the 0.1-1 ms range, which is long compared to the typical inverse frequencies at which the noise current flows in the circuit. Therefore the heat measurements should sample well the occupation probabilities, and below 200 mK this should lead to a negligible poisoning influence, even though poisoning is not completely absent. We have kept the average switching currents figure in the SI to show the crossover to $1e$ periodicity due to thermal QP population.

3. Device limitations/Improvements.

It would be nice if the authors could discuss how this device might be improved, or if its performance (contrast) can be improved. If this picture of gate-modulated low-pass filtering is ok, then one can imagine producing a device with more contrast in the minimum/maximum critical currents. One could also impedance match the lead resistances to the CPT impedance. There's some practical limit to this, but more generally, is there some practical benefit (research or applied) to improving the performance?

There are several ways the device could be improved. As the referee mentions, a better contrast can be achieved by increasing the charging energy: indeed, in strong Coulomb blockade regime (even n_g), the critical current is in good approximation proportional to E_J/E_C . Therefore a higher charging energy increases the effective Josephson inductance and reduces the low-pass cut-off frequency, while it does not (at first order) change the critical current for odd n_g , where the effect of Coulomb blockade is (at first order) cancelled. However, from a fabrication point of view, the permitted range is not so wide: by decreasing, say, the area of the junctions so as to increase E_C , one generally obtains a smaller E_J . While it is not necessarily harmful to the contrast as defined in the manuscript, it leads to a decrease of the maximum thermal conductance (i.e. at odd n_g). One can also play (a little) with the superconducting leads thicknesses: thinner leads increase the BCS gap and thus E_J , while not affecting E_C . Nevertheless, the range of independent tunability remains rather restricted, and there is a loose

trade-off between good contrast and large heat flow for an open heat valve (see related discussion with Referee #3).

On the other hand, indeed, one can work on impedance matching, but here it depends on the kind of performance that one wants. We propose, as a general marker of performance, to introduce the following figure of merit $\beta = G_{max}/G_Q \times (G_{max} - G_{min})/(G_{max} + G_{min})$. It is of interest generally, from a fundamental point of view, if one wants to study quantum effects on heat transport, i.e. to have good matching in some knob position over the full kT/h range (see discussion on the filtering picture above). On the other hand, a good contrast is required to establish qualitatively different regimes of heat transport on a single sample, in the spirit of Ref. [K. Saito & T. Kato PRL **111** 214301 (2013)] (which one would adapt to photons instead of phonons), hence the proposed figure of merit. On the applied aspects, if the goal is to transfer the maximum heat in “open valve” position (i.e., as close as possible to quantum limit of thermal conductance), then it is better to obtain good impedance matching. If the goal is to make a heat switch, regardless of the conduction in open position, then the contrast is more important. Of course, a device combining the two aspects, i.e. having β as close to 1 as possible, is more appealing, e. g. for bolometry applications. Note that temperatures matter here, in that they determine the range of EJ suitable for quantum-limited operation in open valve position.

In order to meet the concerns of Referee #2 and #3, we propose to add a brief discussion on the device optimization before the conclusion.

Summary

This is an interesting paper, although I think that some of the language is a little dramatic for my taste (although I acknowledge that some of this seems necessary for certain journals), and while I'm not suggesting it for publication right away, I'm hoping that my comments/questions can help clarify the picture in the paper so that it's more instructive to a wider audience. I think the bigger questions they're asking are interesting in general, but I think that the experimental discussion could be made clearer.

We hope that the referee finds suitable the revisions made on the manuscript and supplementary material based on his/her useful comments and suggestions.

Reviewer #3 (Remarks to the Author):

In this work, Maillet et al present the realization of heat flow modulation using a fully electric knob, i.e., a Cooper pair transistor. The authors succeed in opening and closing a photonic heat channel between two normal metal reservoirs by playing with the Cooper pair tunneling and Coulomb blockade effect. To do so they use a nice lockin technique to monitor temperature variations, they demonstrate (supp material) that the photonic channel is the main path for heat transport and, finally, they demonstrate that temperature modulation is $2e$ -periodic, univocal proof of the relationship between heat flow and $2e$ quantization in the transistor. There is a sufficient amount of data presented along the paper, figures are extremely simple but really evoking of the phenomenon under investigation. Theoretical analysis of the results is also very clearly explained, rigorous and convincing. To summarize, the work presented by Maillet et al is very beautiful and interesting for a broad scientific community and I recommend its publication in Nat Commun.

We thank the referee for his/her positive assessment of our work, and for the useful comments and questions provided below.

In the following I pose very few questions/suggestions for the authors:

1) In the introduction (abstract) part the authors comment on the flux-charge duality. This statement is confusing to me as they want to avoid any reference to a magnetic flux controlling knob. Would it be better to talk about charge-phase conjugation?

We thank the referee for pointing this, and acknowledge that this statement is confusing. The passage that the referee mentions is indeed relative to charge-phase *conjugation*. We have modified this remark accordingly.

2) Along the paper I didn't get what factors determine the amplitude of the temperature oscillations that the authors are observing. Is it possible to increase this amplitude by, e.g., increasing the thermal gradient or decreasing the cryostat base temperature?

The amplitude of the thermal oscillations is the result of both the amount of AC heating power and the thermal balance. As such, indeed, it depends both on the gradient and the cryostat temperature. However, we stress that we cannot impose too high gradients or too high AC modulations, because this would weaken the linear response approximation that is crucial for quantitative lock-in measurements. On the other hand, decreasing the cryostat temperature will indeed improve the situation in the general case. However, it did not help here because the electronic temperature saturates anyway around 150 mK due to stray radiation, which we think comes primarily from the NIS junctions biasing. Obviously, this limitation is only of technical nature and further setup improvements would help. Overall (see reply to Referee #2), a higher cryostat temperature (in the unsaturated regime, i.e. above 150 mK) means that the electron-phonon relaxation becomes stronger than the photonic one (their magnitudes are already of the same order above 100 mK). As a result, the AC

temperature oscillations at a given n_g are smaller because most of the heat is dissipated to phonons rather than photons. In principle, this does not affect our conductance measurement (which depends only on *the ratio* of the temperature oscillations in source to those in the drain). However, the resolution on the oscillations in each reservoir is poorer if the oscillation amplitudes are smaller, and so is the resolution on the conductance.

We have added a paragraph after Eq. (3), before “Oscillations of the source-drain...” in the main text that details this point, which is also mentioned by Referee #2.

3) I find that the definition of the maximum achieved contrast a bit confusing. Max contrast = 1 means simply that $G_{\gamma, \min} = 0$, am I right?

The referee is correct. We agree that this definition is not perfect, and cannot be considered separately from the ratio between the maximum observed value and the theoretical maximum G_Q . Nevertheless, we consider this definition useful in the sense that an optimized heat valve would indeed allow to have $G_{\min} = 0$ for a specific knob position, which is not the case here. If we consider e.g. our circuit with now smaller and less transparent junctions (higher E_c , smaller EJ), in closed gate position the heat flow should be greatly reduced and the contrast would then approach 1. Yet, the maximum value (open gate) would be smaller than the one we observe because of the EJ reduction. Nevertheless, we believe that with some effort on optimization, it should be possible to win on both sides. Before the main text conclusion, we have added a paragraph that discusses the device optimization in terms of a parameter $\beta = C \times G_{\max}/G_Q$ (C is the contrast as we have defined), which encompasses both the contrast and the maximum achievable heat flow (in open valve position).

4) What do you mean by “negligible retarded response” (methods section) and how do you observe it experimentally?

We refer to the quadrature component of the lock-in response, which accounts for a temperature response to the heater modulation that is retarded. This lag may be due either to finite-frequency filtering in the electrical apparatus or to the finite relaxation time of the heated electrons in the resistor. This relaxation time is the parallel contribution of electron-phonon and electron-photon coupling. At these temperatures the electron-phonon typical relaxation time is in the μs range, i.e. much shorter than the modulation period. Therefore the quadrature component should vanish and indeed we observe a negligible quadrature response (less than 1% of the in-phase signal). In fact, we believe that its non-zero value is primarily due to the line filtering. With an optimized setup AC heating and monitoring of temperature variations at MHz frequencies in the spirit of Ref. [E. Pinsolle *et al.*, PRL **116** 236601 (2016)] may give valuable insights on heat relaxation mechanisms.

We have re-written the corresponding statement in the Methods section with more details.

5) What do you mean by “the issue of finite frequency behavior” (conclusions section)?

The measurements presented in this paper are only performed in a DC (or quasi-DC) regime, and as a result the quantities that are measured may be seen as the zero-frequency limit of the resistors voltage

noise spectral density. Therefore, we lack the frequency content of those spectra, and we do not have access to correlations between (voltage) noise in resistor 1 and noise in resistor 2. It has been shown in prior works, both theoretically [D. S. Golubev *et al.*, PRB **92** 085412 (2015)] and experimentally in a purely classical regime [S. Ciliberto *et al.*, PRL **110** 181601 (2013)], that correlations carry thermodynamic information on the non-equilibrium steady state resulting from the temperature gradient. In particular a time-resolved analysis of voltage fluctuations may for instance yield the entropy produced over time over a single thermodynamic trajectory, and could allow one to investigate what entropy production and the second law of thermodynamics become in the quantum regime (i.e. when the heat flow is limited by either $k_B T/\hbar$ or by a quantum circuit), both on average and at the statistical level.

6) Finally, I have a fundamental question. What is exactly the role played by the superconductor? As far as I understand it, the role of the superconducting part is to block heat currents mediated by quasiparticles. If this is the case, could it be possible to demonstrate a similar effect using a conventional normal-metal single-electron transistor stacked between superconducting leads or am I missing something? Is that the equivalent effect that you are observing at large temperatures where you attribute the 1e-periodicity to quasiparticle poisoning (supp material)? If this is the case, has this effect been observed experimentally previously?

The referee raises a valid point, which is rather subtle to address. The superconducting leads are indeed primarily intended for quasiparticle heat current suppression (also, normal leads would be resistive and contribute to impedance mismatch). Fundamentally, a normal SET inclusion may be indeed used as a photonic heat valve, because it is also to some extent a gate-tunable impedance (see discussion below).

However, the physical details of what happens at the valve level in a normal SET case differ significantly from the fully superconducting case. A normal SET transports only unpaired electrons that dissipatively tunnel: after a tunneling event, an electron equilibrates with the bath through fast (ns range) electron-electron scattering. Under zero net bias applied to the SET this translates in equivalent electrical terms as a zero-bias DC conductance which varies between at best $1/2R_{SET}$ (gate open, R_{SET} is the SET series tunnel resistance at large voltages) and worst roughly $1/R_{SET} \times e^{-E_c/k_B T}$ (gate closed, E_c is the island's charging energy of order 1K). Thus, it behaves as an effective RC circuit ($C \sim 0.1 - 1$ fF is the junctions series capacitance), which is indeed electrically tunable with the gate voltage but very lossy, as SETs have typical tunnel resistances bigger than $\hbar/e^2 \approx 25$ k Ω , which is already far bigger than the typical resistances R_i that we implement in our experiment (below 1 k Ω). Therefore, even in the "loop" configuration that we use, the transmission coefficient detailed in the paper would be much smaller than 1 (at best roughly $4R_i^2/R_{SET}^2 \sim 10^{-5}$) in the "thermal" frequency range (at best, the high-pass filter cut-off $1/RC \sim 10$ GHz $> kT/\hbar$ at our temperatures) due to this large impedance mismatch. As a result the photon heat flow would probably be unobservable by our means.

As an opening, one can wonder what happens in terms of shot noise, which is disregarded in the above explanation but nonetheless relevant: in a recent preprint [S. Larocque *et al.*, ArXiv 2002.10339 (2020)] the shot noise of a normal tunnel junction under a temperature bias was measured. In view of these results, one can imagine that a temperature-biased normal SET would act as a "secondary" noise current source. The impact on the thermal balance as well as the influence of Coulomb blockade may be then interesting to investigate.

Physically, the superconducting SET under zero bias acts very differently: its zero bias impedance can be, ideally, approximated purely as a reactance (dominated here by the Josephson inductance, see Referee #2 comment and reply) because the Cooper pairs that carry the current noise due to the thermal gradient tunnel non-dissipatively via the Josephson coupling. The subgap resistance of the superconducting SET is irrelevant here as it only concerns quasiparticles and in fact, participates further to suppress any hypothetical electronic heat flow (see SuppMat).

In summary, both strategies are possible, but the normal SET seems rather unpractical in our implementation. The fundamental difference between the two is the nature of the current noise carriers (Cooper pairs and Josephson tunneling versus normal electrons and dissipative tunneling), which leads to a very different transmission behaviour for electromagnetic heat in the two cases.

As for the $1e$ periodicity observed at high temperatures, this should indeed translate as $1e$ periodicity in the photonic thermal conductance. However, such effect is in practice not observed, because the $1e$ modulation is very weak in contrast, while electron-phonon relaxation overwhelms electron-photon relaxation at corresponding temperatures: therefore, temperature oscillations both in source and drain are small, and as a result the noise on conductance is too large to see the modulation. We did, however, observe a weak modulation of the conductance reappearing above 300 mK, but this is likely due to *electronic* heat flow rather than *photonic*: this predominance at such temperatures was already observed in Ref. [A. V. Timofeev *et al.*, PRL **102** 200801 (2009)], due to a non-negligible quasiparticle population in the superconducting leads. Such a situation is closer, in an imperfect way, to the experiment described in Ref. [B. Dutta *et al.*, PRL **119** 077701 (2017)]. Even if this situation is itself interesting, we did not include these results in the main text or the SI as we felt they were out of the scope of the paper.

REVIEWERS' COMMENTS:

Reviewer #1 (Remarks to the Author):

The authors' replies and the corresponding revisions seem satisfactory. I would recommend the revised manuscript for publication in its current form.

Reviewer #2 (Remarks to the Author):

I have read the authors' response to my initial review (as well as the others). They have written a thoughtful response and made adjustments to the text that address all of my comments.

I'm happy to suggest this for publication.

Reviewer #3 (Remarks to the Author):

Thank you very much for the clarifications, very nice work!

Maria Jose Martinez-Perez

REVIEWERS' COMMENTS:

Reviewer #1 (Remarks to the Author):

The authors' replies and the corresponding revisions seem satisfactory. I would recommend the revised manuscript for publication in its current form.

We thank the referee for the positive appreciation and recommendation.

Reviewer #2 (Remarks to the Author):

I have read the authors' response to my initial review (as well as the others). They have written a thoughtful response and made adjustments to the text that address all of my comments.

I'm happy to suggest this for publication.

We thank the referee for the positive appreciation and suggestion.

Reviewer #3 (Remarks to the Author):

Thank you very much for the clarifications, very nice work!

Maria Jose Martinez-Perez

We thank Dr. Martinez-Perez for her positive appreciation and kind comment.